# Acquired Male Hypogonadism in the Post-Genomic Era—A Narrative Review

**DOI:** 10.3390/life13091854

**Published:** 2023-09-01

**Authors:** Giuseppe Grande, Andrea Graziani, Luca De Toni, Andrea Garolla, Domenico Milardi, Alberto Ferlin

**Affiliations:** 1Unit of Andrology and Reproductive Medicine, Department of Medicine, University of Padova, 35128 Padova, Italy; ccgmpadova@gmail.com (A.G.); segreteria.endocrinologia@rm.unicatt.it (L.D.T.); andrea.garolla@unipd.it (A.G.); alberto.ferlin@unipd.it (A.F.); 2Division of Endocrinology, Fondazione Policlinico Universitario “Agostino Gemelli” Scientific Hospitalization and Treatment Institute (IRCCS), 00168 Rome, Italy; domenico.milardi@policlinicogemelli.it

**Keywords:** testosterone, hypogonadism, proteomics, metabolomics, markers

## Abstract

Although precision medicine took its first steps from genomic medicine, it has gone far beyond genomics, considering the full complexity of cellular physiology. Therefore, the present time can be considered as the “post-genomic era”. In detail, proteomics captures the overall protein profile of an analyzed sample, whilst metabolomics has the purpose of studying the molecular aspects of a known medical condition through the measurement of metabolites with low molecular weight in biological specimens. In this review, the role of post-genomic platforms, namely proteomics and metabolomics, is evaluated with a specific interest in their application for the identification of novel biomarkers in male hypogonadism and in the identification of new perspectives of knowledge on the pathophysiological function of testosterone. Post-genomic platforms, including MS-based proteomics and metabolomics based on ultra-high-performance liquid chromatography-HRMS, have been applied to find solutions to clinical questions related to the diagnosis and treatment of male hypogonadism. In detail, seminal proteomics helped us in identifying novel non-invasive markers of androgen activity to be translated into clinical practice, sperm proteomics revealed the role of testosterone in spermatogenesis, while serum metabolomics helped identify the different metabolic pathways associated with testosterone deficiency and replacement treatment, both in patients with insulin sensitivity and patients with insulin resistance.

## 1. Introduction

Male hypogonadism (MH) has been defined as “a clinical syndrome that results from failure of the testes to produce physiological concentrations of testosterone (T) (T deficiency) and/or a normal number of spermatozoa due to pathology at one or more concentrations of the hypothalamic–pituitary–testicular (HPT) axis” [1]. According to the site involved, there are two main forms of MH: primary hypogonadism (or hypergonadotropic hypogonadism), due to a primitive testicular alteration, and secondary hypogonadism (or hypogonadotropic hypogonadism (HH)), caused by a disease of the pituitary gland. In the first case, patients have low/normal T concentrations and high FSH and LH concentrations whilst, in the latter, patients present low T concentrations in association with normal or low FSH and LH concentrations.

Primary MH is mainly caused, among other causes, by Klinefelter syndrome, cryptorchidism, chemotherapy, radiation to the testes or the pelvic area, trauma and testicular torsion [1]. Secondary MH, on the other hand, is mainly caused by hyperprolactinemia, obesity and metabolic alterations, the use of opioids, glucocorticoids or androgen-deprivation therapy, hypothalamic or pituitary oncological or infiltrative disease, traumas and pituitary radiation or neurosurgery [1].

Some authors classify hypogonadism based on the onset of symptoms or the reversibility of the clinical condition. The term “organic hypogonadism” identifies a type of hypogonadism that is not reversible, due to congenital or acquired alteration at the hypothalamic–pituitary–testicular (HPT) axis [2]. Otherwise, the term “functional hypogonadism” (FH) is referred to a potentially treatable condition of the HPT feedback loop. Most patients with FH are middle-aged or older men. In fact, FH is associated with comorbid illnesses which might eventually lower the reduction in T concentrations [3,4]. In addition, unlike classical MH, FH usually is characterized by a more subtle and non-specific clinical condition, with the absence of signs and symptoms specific to low T concentrations [5]. In most men affected by FH, gonadotropin concentrations are not elevated as might be expected; in fact, this type of hypogonadism is usually caused by functional HPT axis suppression with an intact HPT axis, a picture which is similar to functional amenorrhea in women [3]. 

The diagnostic flowchart and evaluation of men with suspected hypogonadism include medical history, physical examination and hormonal and instrumental evaluation. Basal evaluation of serum total testosterone (TT), sex hormone binding globulin (SHBG), albumin, LH and FSH concentrations represents the first step in order to evaluate the presence of T deficiency. If reduced T levels are associated with low or low-normal gonadotropin levels, a magnetic resonance imaging scan of the pituitary gland is mandatory, in order to study the sellar and parasellar region, besides a biochemical evaluation of the pituitary function (Lenzi [5]).

Total T levels < 8 nmol/L are strongly associated with the diagnosis of MH. A condition of hypogonadism is otherwise suspected in cases of total T concentrations between 8 and 12 nmol/L. It is a “gray zone” in which the diagnosis of MH is not simple, and the indication to treatment should be defined by the presence of signs/symptoms of hypogonadism, although frequently, those are not specific [6,7].

In all patients, but especially in men with testosterone levels in the “gray zone” and sexual symptoms of MH, calculated free testosterone (cFT) determination, obtained by dosing testosterone, albumin and SHBG levels, allows a more accurate diagnosis of hypogonadism, since total testosterone determination alone misdiagnoses hypogonadism in about 8% of men with sexual symptoms [8].

Both low levels of TT and/or cFT and the presence of signs/symptoms of MH are required to diagnose MH [1,2]. Furthermore, routine screening of MH in the general population is not recommended in the absence of known risk factors of MH and/or the presence of peculiar signs/symptoms [5]. In fact, low T concentrations occur frequently without symptoms/signs of MH, without establishing a diagnosis of hypogonadism.

Low TT concentrations are defined when levels are below 12.0 nmol/L, whilst low cFT concentrations are defined in the presence of cFT levels below 220 pmol/L. In addition, a concentration of LH above 9.4 IU/L, in the presence of low TT/cFT, suggests primary hypogonadism [2]. 

Clinicians should also bear in mind that TT concentrations should be measured on two separate mornings when the patient is fasting, before 10–11 am [1,2,5]. Moreover, signs and symptoms of MH are not specific and accurate indicators of this disease. In particular, they include, among others, small testes, reduced libido, decreased spontaneous erection and erectile dysfunction, gynecomastia, hot flushes and sweats, osteoporosis and low bone mineral density, infertility, sperm count alterations, poor concentration and memory, depression, obesity, reduced body hair and reduced muscle strength [1,2,5].

In light of the evidence exposed above, testosterone replacement therapy (TRT) is recommended in men with low TT/cFT concentrations and signs/symptoms of hypogonadism [1,2]. Furthermore, lifestyle changes, including physical exercise and weight reduction, and withdrawal/modification of drugs potentially interfering with T production are recommended [5]. Regarding lifestyle changes, for instance, it is known that obesity is a strong risk factor associated with low T concentrations. A weight reduction of >5% causes, in fact, an increase in testosterone concentration of 2 nmol/L [3]. 

There are different testosterone formulations that are approved as TRT, including intramuscular administration, transdermal gel and patch administration and injectable long-acting testosterone administration [1,2]. Contraindications to TRT include active breast and prostate cancers, recent major adverse cardiovascular events, a high risk of a thromboembolic event, severe and untreated obstructive sleep apnea syndrome, desire of fatherhood, high levels or increasing levels of hematocrit and/or prostate specific-antigen and severe hepatic dysfunctions. 

The aim of this manuscript is therefore to evaluate and delve into the role of post-genomic platforms, i.e., proteomics and metabolomics, with a specific interest in their application for the identification of novel biomarkers in MH and their application for new perspectives of knowledge on the pathophysiological function of testosterone.

## 2. Personalized -Omic Medicine in Post-Genomic Era

The term “-omics” deals with the study of biological systems on a large scale.

According to the incorporation of Mendelian genetics into medicine, non-infectious diseases are associated with alterations or mutations in specific genes. Because of those alterations, a known clinical condition might have the same treatment in any presenting patient. This “one disease–one treatment” dualism was put into discussion by observations of the variable responses to medications. Moreover, the influence of epigenetic and environmental factors was reported to be associated with different clinical presentations. These clinical conditions are considered as multifactorial diseases, and their management underlines the necessity of analyzing the impact of genetics, epigenetics and the environment on disease course, from the perspective of a personalized treatment [9].

Regarding the evidence exposed above, the first step was the Human Genome Project, in 2003, which allowed the identification of single-nucleotide polymorphisms (SNPs) [9]. SNPs explain 90% of genetic polymorphisms. Characterization of SNPs in clinical practice might represent the molecular signature associated with diagnosis, prognosis and therapy, from the perspective of personalized medicine [10]. Further technological progress has plunged the cost of sequencing the human genome [11]. 

Notwithstanding the level of detail provided by a genome sequence, it must be kept in mind that a genome sequence defines just one element involved in multifactorial diseases. Thus, the Human Genome Project discovered more than 20,000 protein-coding genes (~3% of the genome). However, further studies revealed several regulation mechanisms associated with non-coding DNA sequences [12].

Genetics alone is therefore not able to predict the protein patterns associated with post-translational modifications (PTMs) or protein–protein interactions.

So, although precision medicine took its first steps from genomic medicine, it has gone far beyond genomics, considering the complexity of cellular physiology [9]. Therefore, the present time can be considered as the “post-genomic era”. 

It is possible to define the “proteome” as the total amount of proteins present in a cell at a given time, which depends on their localization, interactions, post-translational modifications and turnover. Wilkins used the word “proteomics” to describe the “PROTein complement of a genOME” in 1996 [13]. Instead of capturing the expression of specific genes, proteomics catches the overall protein profile. Only proteome analysis can unravel personalized medicine’s biological complexity. Consequently, proteomic- and interactomic-based individual care is flexible, adjusting to people and their circumstances. Therefore, the most comprehensive, integrated platform for the clinical analysis of a clinical specimen is mass spectrometry (MS)-based proteomics. Furthermore, recent technological and bioinformatic innovations permitted the definition of the specific molecular signatures of several clinical conditions [14], thus permitting proteomics to provide a genuine guarantee in translational investigation for early diagnosis, prognosis and theragnosis on an individual basis. In the area of male reproduction, proteomics represents the most promising and powerful platform in order to widely study the physiology and pathophysiology of male reproduction and to identify novel markers of disease [15,16].

A new type of science, known as metabolomics, aims to study the molecular fingerprints of a given clinical condition, by the high-throughput measurement of intermediate or end-point metabolites in a biological sample. Metabolomics is considered the endpoint of the “-omic” cascade [17]. Because of the high sensitivity of metabolomics, specific alterations in biological pathways can be identified to provide insight into the pathophysiological mechanisms involved in diseases [18].

In the area of andrology and in the clinical workflow of hypogonadic patients, the introduction of evidence derived from post-omic sciences might permit andrology to enter a new -omic era, from the perspective of personalized and individualized care. In fact, current clinical guidelines on this topic acknowledge important gaps in the current evidence base that have led to important differences in interpretation and approaches for optimizing individualized management of patients, especially for older men with non-classical hypogonadism [1]. Starting from these premises, new markers of androgen activity in these patients might be implemented in future specific recommendations in practice, in order to optimize benefits and minimize risks for patients, from the perspective of personalized medicine.

In this review, the role of post-genomic platforms, namely proteomics and metabolomics, is analyzed with specific insight into their application in the identification of novel biomarkers in MH and new knowledge on the mechanism of T action.

## 3. Seminal Proteomics in MH: Revealing Peripheral Markers of T Action

The action of a clinical picture of MH and thus the role of testosterone replacement treatment (TRT) on the function of male accessory sex glands have been widely reported and demonstrated. A study by Brooks showed that, in rats, orchiectomy causes a weight reduction of accessory glands [19]. A high androgen sensitivity is widely known for prostate. In particular, androgens are crucial for prostate development, growth and functions. In addition, signaling via the AR axis is important in facilitating prostate carcinogenesis; in fact, the risk and outcome of prostate cancer have been associated with serum TT concentrations and polymorphisms in the androgen signaling pathway [20]. In 2009, Ma et al. reported that 187 transcripts were reduced in the prostatic tissue of mice with dihydrotestosterone reduction [21], confirming that androgen-modulated genes are essential for prostate function.

In 2014, we successfully proposed the first experimental proteomic original study, utilizing high-resolution mass spectrometry, aimed at evaluating the seminal proteome of hypogonadic patients, at admission and after 6 months of TRT [22]. Semen proteomic analysis was performed in 20 male patients with HH. Ten normogonadic subjects were considered as a control gather. Seminal plasma proteomic assay was performed using an Ultimate 3000 Nano/Micro-HPLC apparatus equipped with an FLM-3000-Flow manager module and coupled with an LTQ Orbitrap XL hybrid mass spectrometer. Sixty-one proteins were recognized within the gather of prolific men. Thirty-three of these identified proteins were missing in all the seminal samples from patients with MH. Fourteen out of thirty-three missing proteins were found in the seminal samples of hypogonadic patients after 6 months of TRT, and seven out of fourteen differentially expressed proteins (prolactin-inducible proteins, lactoferrin, prostatic acid phosphatase, myeloperoxidase; zinc-alpha-2-glycoprotein, lactotransferrin, cystatin C) fell in a functional protein-protein network, centered on AR.

We later repeated the study by using a more modern technological platform and a quantitative bioinformatic strategy [23] and studying 10 male patients with postsurgical HH. Five patients were studied after a brief course of 3 months of therapy with transdermal T to analyze the effect of T therapy on the seminal proteomic profile. The evaluation after only 3 months of TRT provided us with several pieces of information. A 3-month period might be the correct timing of response to exogenous T [24]. After the digestion of proteins and mass spectrometry analysis, the quantitative analysis was performed, as reported in Figure 1. Eleven proteins were found to be lowered in the patients’ group. In the population of five patients treated with T therapy, we defined a list of under-expressed (ratio > 1.5) or overexpressed (ratio < 0.67) proteins after that therapy. Five proteins (semenogelin 1, semenogelin 2, prolactin-inducible protein, prostatic acid phosphatase, lactotransferrin) increased after TRT, as reported in Table 1. Finally, Western blot analysis confirmed proteomic data. 

Semenogelins are produced by seminal vesicles and provide the structural components for coagulum formation after ejaculation. They bind with eppin, and their major functions are involved in controlling the motility of sperm and capacitation and in transporting the immune-modulating activity for the sperm during the transit in the female reproductive tract [25]. 

Prolactine-inducible protein (PIP) is an aspartyl proteinase that is tied to various proteins, including human zinc-α-2 glycoprotein [26]. It has been detailed that PIP is involved in fertility, antimicrobial activity and immune regulation [27]. A reduction in seminal PIP was additionally depicted in asthenospermic patients [28]. 

Prostatic acid phosphatase (PPAP) is a binding protein for many prostatic proteins and is involved in the restraint of sperm motility and in avoiding spontaneous acrosome reactions in sperm [29]. 

Finally, lactotrasferrin (TRFL) has an antioxidative, antibacterial and immune-modulating role in seminal plasma [30]. Lactoferrin is additionally involved in maintaining sperm structure and motility and in tweaking the composition and quality of the semen during sperm maturation and migration through the genital tract [31].

Although further studies are needed in order to understand whether testing this group of proteins might be translated into clinical practice, there is no doubt that, if confirmed in big populations of hypogonadic patients, this panel might help in the diagnosis and follow-up of patients with MH, specifically in older patients with mild symptoms and T concentrations in the “gray zone”, to differentiate the physiological mild decline in T levels from a reduction in T action, the latter needing the start of TRT. It is probably time, in fact, to move to a “molecular androtest”, representing a seminal fingerprint of male T peripheral action. From this perspective, seminal proteomics provided us with the most promising platform for this aim [6]. 

## 4. Sperm Proteomics in Hypogonadism: Unrevealing the Role of Testosterone in Spermatogenesis

High-resolution sperm proteomics in patients affected by MH has been used to study the role of T in spermatogenesis [32]. 

It is in fact well-studied that T is necessary in order to both start and maintain the development of sperm cells and that the production of these cells depends on androgen action within the testis. Mice that lack Sertoli cell androgen receptors were reported to be affected by late meiotic germ cell arrest, indicating that in Sertoli cells, androgens might regulate this crucial phase in spermatogenesis. Furthermore, any change in this hormonal pathway may impact molecules involved in the meiotic division, cell death, oxidative damage, DNA repair and RNA processing [33]. 

In light of this evidence, sperm proteomics has been performed in a rare clinical condition represented by LH deficiency, in which a dysregulation in the HPT axis resulted in an alteration of LH secretion, although normal FSH production was still maintained [34]. This model represented a clinical picture of a severe lowering in both blood and intratesticular T concentrations, in the absence of any other confounding risk factors. In this rare and particular clinical model, both testosterone and LH deficiencies are associated with normal or low-normal FSH production, further associated with the presence of spermatogenesis. A comparative proteomic evaluation was performed, comparing sperm from controls (five fertile men with normal hormonal parameters and normal semen analysis) versus sperm from patients with secondary hypogonadism (five patients affected by isolated LH deficiency and oligozoospermia). Sperm protein composition was analyzed using liquid chromatography followed by tandem mass spectrometry (LC-MS/MS) on an LTQ Velos-Orbitrap mass spectrometer. Bioinformatic tools were used for high-reliability protein identification. The methodological approach is shown in Figure 2. A total of 986 proteins were found, and for 43 among them, a differential expression was observed: 32 proteins were reduced and 11 were increased in hypogonadic patients versus controls. In particular, 13 proteins have been previously described as involved in sperm function and spermatogenesis. 

In detail, the study underlined the role of T deficiency in inducing a dysregulation of the molecular mechanisms associated with spermiogenesis, the blood–testis barrier, spermiation and the epididymal shedding of cytoplasmic droplets, as documented by the reduction in four proteins (Prosaposin, Ropporin-1B, Plasma serine protease inhibitor and SPARC-related modular calcium-binding protein 1). 

Prosaposin is a 65–70 kilodalton protein, also known as SGP-1 for sulfated glyprotein-1. It is produced in many tissues from a highly conserved gene. Normally, prosaposins are targeted to lysosomes and processed into four biologically active saposins (each 7–10 kd, called saposins A–D) plus three residual peptide chains of 15 to 50 amino acids [35]. Prosaposin is one of the major glycoproteins secreted by rat Sertoli cells in the tubular lumen [36]. In an immunocytochemical study using a gold-labeled SGP-1 antibody, it was observed that the Sertoli cell lysosomes were immunoreactive [37,38]. In these organelles, prosaposins, presumably through their saposin derivatives, can play a role in the lysis of the Sertoli cell plasma membrane, which is internalized through a particularly active endocytosis during stages IX to XIV of the cycle of the seminiferous epithelium [39,40]. Prosoposin has also been implicated in the degradation of the residual bodies released by the spermatids during spermiation [41,42]. Prosaposin binds the GPR37 receptor. Although Gpr37−/− adult mice were fertile, they exhibited reduced testicular weight, lower epididymal sperm counts and defects in sperm motility [43]. Prosaposin is moreover secreted by epididymal principal cells [41,44], and in roosters, it was demonstrated that a fragment of prosaposin promotes sperm–egg binding [45].

Ropporin is a sperm-specific binding protein for rhophilin that is located in the fibrous coat of the spermatic flagellum. Previous studies have demonstrated that ropporin is indeed concentrated in the principal piece and the end piece of the spermatic flagellum [46]. The expression level of ropporin was significantly lower in normal controls [47], suggesting that ropporin may be related to sperm motility, and its reduced expression may contribute to the reduced sperm motility in asthenozoospermic patients. Furthermore, in infertile asthenozoospermic men with varicocele, ropporin gene expression in sperm cells increased after varicocelectomy and was correlated with improved sperm parameters and decreased vein diameters [48]. 

Plasma serine protease inhibitor, also designated as Protein C inhibitor (PCI) or Serpin A5, is a member of the serine protease inhibitor (serpin) family that inactivates serine proteases by forming a stable, enzymatically inactive 1:1 enzyme inhibitor complex [49]. PCI inhibits many proteases such as activated protein C [50,51], thrombin [52], factor Xa [52], factor Xia [52,53], plasma kallikrein [52,53], thrombin–thrombomodulin complex [54], urokinase (uPA) [55,56], tissue plasminogen activator (tPA) [52,56], the sperm protease acrosin [57,58], tissue kallikrein [59,60] and prostate specific antigen (PSA) [60,61]. Disruption of the PCI gene, which is highly reported in the male reproductive tract, leads to infertility in homozygous PCI-knockout male mice [62]. Sperm derived from PCI−/− males is malformed and unable to bind and fertilize oocytes, as demonstrated by in vivo and in vitro fertilization experiments. Histologic analysis of PCI−/− mice revealed abnormal spermatogenesis associated with damage to Sertoli cells, possibly due to unhindered proteolytic activity [62]. Besides the importance of PCI in spermatogenesis and spermiogenesis, previous immunohistochemistry studies, as well as sperm malformations observed in PCI−/− deficient mice, suggest that PCI may play a role in further sperm cell maturation during the passage during epididymal migration, as inferred from the presence of PCI in the cytoplasmic droplet [49]. In fact, after acrosome formation is completed, the Golgi apparatus migrates to the proximal part of the middle piece and the cytoplasmic protrusion containing the remaining Golgi stacks forming the cytoplasmic droplet. During the passage of spermatozoa to the cauda epididymidis, the cytoplasmic droplets exhibit typical migration from the proximal middle piece of spermatozoa to the tail. The cytoplasmic droplets are eventually eliminated and are thought to have been digested by epithelial cells of the epididymidal duct. Spermatozoa that fail to shed cytoplasmic droplets display immotile, bent flagella as observed in PCI−/− mice [62]. The presence of cytoplasmic droplets in ejaculated sperm is used as a marker of infertility, underscoring the fact that cytoplasmic droplets are important for sperm maturation.

SMOC1 (SPARC related modular calcium binding 1) is a matrix protein that belongs to the SPARC (Secreted acidic cysteine rich glycoprotein; also known as BM-40/osteonectin) protein family. Matricellular proteins influence many cellular functions including growth factor signaling, cell migration, cell adhesion and proliferation [63]. SPARC is the best-characterized member of this class and has been shown to interfere with the receptor-mediated signaling of many growth factors including PDGF [64], VEGF [65], FGF2 [66] and IGF1 [67]. In the testis, SMOC1 is involved in intercellular signaling and cell type-specific differentiation during gonadal and reproductive tract development, participating in the differentiation of the supporting cell lineage and in the interactions between Sertoli and germ cells [68].

In summary, differential expression of these proteins in sperm may reflect dysregulation in the MH of the molecular machinery involved in spermatogenesis, Sertoli cell junctions and the blood–testis barrier, spermiation and the epididymal shedding of cytoplasmic droplets.

In addition, overexpression of Sperm protein associated with the nucleus on the X chromosome B1 (SPANXB1) and Germ cell-specific gene 1 protein (GSG1) proteins, previously described to be associated with defective spermatogenesis [69,70,71], and Epididymal sperm-binding protein 1 (ELSPBP1) and fibronectin (FN1), markers of sperm damage and low sperm quality [72,73], underlines the weight of T in the qualitative control of sperm maturation. 

Finally, the overexpression of 5-oxoprolinase underlined the role of T deficiency in sperm oxidative stress [74]. 

Overall, these proteomic data revealed the physiological role of intratesticular T in regulating the expression of molecular machinery important for spermatogenesis and spermiogenesis, spermiation, sperm quality control, sperm motility and sperm oxidative stress. 

As a consequence, modern proteomic and bioinformatic platforms are permitted to re-evaluate the role of T in spermatogenesis and in sperm function.

## 5. Serum Metabolomics in Hypogonadism: Metabolic Differences between Functional and Primary Hypogonadism

T has a marked influence on the adipose distribution of the body and on the preservation of bone and muscle mass. In particular, T therapy may inhibit bone resorption and increase bone mass at the lumbar spine level [75], further having a noteworthy role in glucose homeostasis and lipid metabolism [76]. Moreover, as described by Foresta et al. [76], sexual symptoms and alteration of testosterone concentrations might represent a harbinger for further cardiovascular and metabolic investigation. Furthermore, MH induces metabolic alterations through several molecular and physiological pathways [77,78,79]. In light of this evidence, a hypogonadal state might be present, in normoinsulinaemic patients (insulin-sensitive (IS) patients) not at the beginning, leading over time to an increase in blood insulin concentrations, metabolic dysfunction and clinical complications. In fact, MH is strongly associated with visceral adiposity, reduction in insulin sensitivity, alteration of glucose tolerance, elevated triglyceride concentrations and, in general, alteration of the lipidic profile. 

Furthermore, hypogonadism might occur in the absence of structural alterations of the HPT axis and pathologic conditions that suppress the HPT axis. This condition, previously presented as FH, is thought to be caused by metabolic alterations, including obesity and type 2 diabetes mellitus [2,4,5]. Previous population studies reported that obesity and weight gain increased the risk of developing FH in the group under study. In contrast, in men who encountered weight loss during the follow-up period, T concentrations often increased after the restoration of eugonadism [80,81]. The mechanisms underlying those functional alterations have not been completely clarified yet. However, a possible explanation might be associated with metabolic disturbances and inflammation which are linked with obesity, further interfering with T/gonadotropin production at several levels [82,83]. 

In light of this evidence, metabolomic science aided in revealing the several metabolic pathways associated with primary and functional hypogonadism and described how TRT might indeed change these dysmetabolic pathways.

In 2018, Fanelli et al. carried out a serum metabolomic study using ultra-high-performance liquid chromatography-HRMS. The studied population included 15 patients affected by MH and with normal insulin sensitivity, compared with 15 controls with normal hormonal parameters [84]. Further, the same study evaluated the metabolomic profiles of men affected by hypogonadism and IS, after 60 days of the administration of transdermal testosterone [85]. The same studies have been repeated in patients with functional hypogonadism, characterized by the presence of obesity and insulin resistance, before [86] and after 60 days of TRT [87]. 

Taking these studies together, we might analyze the differentially expressed metabolic pathways, as reported in Table 2. 

In patients with primary hypogonadism, glycolysis is usually normal, indicating that, in this subtype of patients, glucose is used in muscle, adipose and liver as the main biofuel, whilst alternative sources of energy are minimally used. Amino acids are not employed for energy production [84]. After TRT, a reactivated and accelerated glycolysis (as confirmed by lactate increase) represents the principal source of energy in these patients [85]. Conversely, in functional hypogonadism, glycolysis is the most consistently altered biochemical process in IR patients, and glucose metabolism is fuelled by amino acid degradation, which is highly increased [86]. After TRT, there is a weak improvement in glycolysis, so lactate is higher after TRT [87]. Gluconeogenesis is inactive in patients with primary hypogonadism [84] but strongly activated in patients with functional hypogonadism [86] and stopped in these patients after TRT treatment [87].

Moreover, TRT is associated, in patients with primary hypogonadism, with decreased levels of free carnitine, so it has been suggested that the integration of carnitine, in association with TRT, might be useful in these patients [84]. 

Conversely, TRT was associated in patients with functional hypogonadism with the development of ketone bodies. Indeed, symptoms often described after the initiation of TRT (concentration difficulties, depression, brain fog, memory impairment) may be related to the metabolic ketonic state. This led to the proposal of the existence of IR patients affected by hypogonadism receiving TRT of a specific “keto flu-like” syndrome.

In conclusion, metabolomic studies provide new perspectives for the diagnosis (i.e., differentiation of primary and functional hypogonadism), follow-up (i.e., assay of ketone bodies after TRT in patients with functional hypogonadism) and treatment (i.e., integration of carnitine in association with TRT in patients with primary hypogonadism) of patients with primary and functional hypogonadism.

## 6. Conclusions

High-throughput -omics technologies let us study a large number of omics markers at the same time. Post-genomic platforms, including MS-based proteomics and ultra-high-performance liquid chromatography-HRMS-based metabolomics, have been applied to respond to clinical questions associated with MH and to define new markers of MH to be translated into clinical practice, both related to reproduction and metabolism. In detail, seminal proteomics helped us in identifying novel non-invasive markers of androgen activity to be translated into clinical practice, sperm proteomics revealed the role of testosterone in spermatogenesis, while serum metabolomics helped identify the different metabolic pathways involved in MH and TRT, in the case of both primary and functional MH.

Proteomic results will be useful in the diagnosis and follow-up of patients with male hypogonadism, specifically in patients with mild symptoms and T concentrations in the “gray zone”, in order to move to a “molecular androtest”, representing a seminal fingerprint of male T peripheral action. Metabolomics will also be useful both for the follow-up and the improvement of therapeutical protocols in patients with MH.

The challenge for the future is therefore represented, also in the field of MH, in combining multi-omics data to highlight the interrelationships of the involved biomolecules for disease subtyping (i.e., functional vs. primary hypogonadism), biomarker prediction (i.e., markers of hypogonadism) and obtaining prognostic and therapeutic advancements from the perspective of personalized medicine [89].

## Figures and Tables

**Figure 1 life-13-01854-f001:**
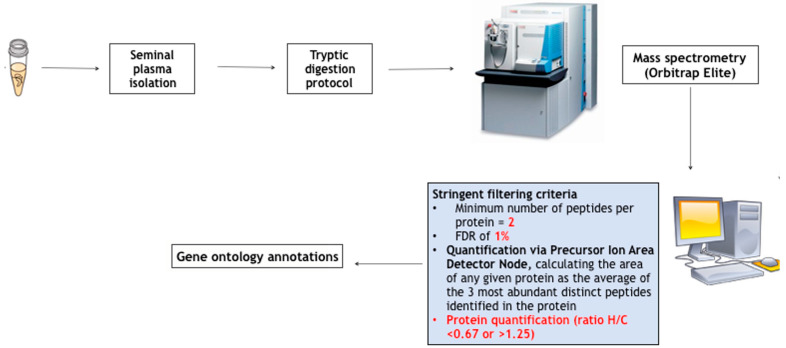
Methodological process of seminal plasma proteomics and bioinformatic analysis.

**Figure 2 life-13-01854-f002:**
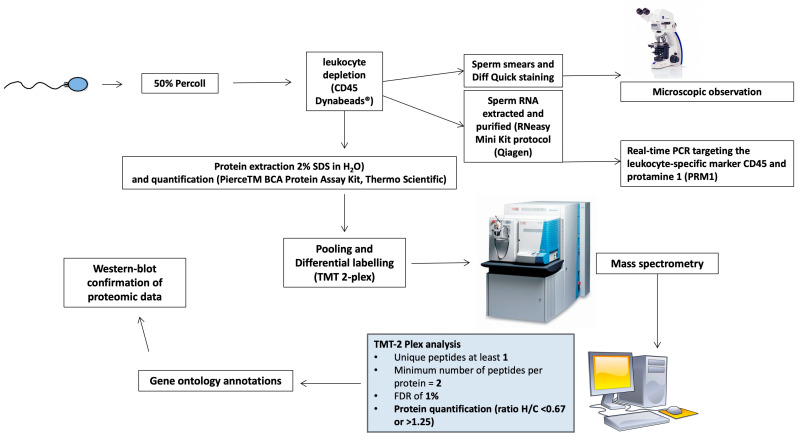
Methodological process of sperm proteomics and bioinformatic analysis in hypogonadic patients and controls.

**Table 1 life-13-01854-t001:** Panel of putative seminal makers from proteomic evidence of MH [6].

Protein	Gene
Semenogelin-1	SEMG1
Semenogelin-2	SEMG2
Prolactine-inducible protein	PIP
Prostatic acid phosphatase	PPAP
Lactotransferrin	TRFL

**Table 2 life-13-01854-t002:** Major reports of metabolomic studies in MH. Abbreviations: TRT: testosterone replacement therapy [88].

	Primary Hypogonadism		Functional Hypogonadism	
Metabolic Pathway	Before TRT	After TRT	Before TRT	After TRT
Glycolysis	decreased	increased	highly decreased	increased
Gluconeogenesis	inactive	inactive	increased	decreased
Ketone body formation	inactive		increased	highly increased

## Data Availability

Not applicable.

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
