# Peer review of "Acquired Male Hypogonadism in the Post-Genomic Era—A Narrative Review"

_life, 2023, doi:10.3390/life13091854_

Round 1
Reviewer 1 Report
The manuscript is very interesting, well organized and well written. The topic is original and innovative.
Even if it is already clear, I suggest to describe “the aim” of the review at the end of the "introduction" paragraph.
Furthermore, to standardize the paragraphs, I suggest to remove the subtitle from paragraph 2.
Finally, I suggest defining the acronyms in the table legends.
Good
Author Response
We are grateful to the Reviewer for his/her observations.
The manuscript is very interesting, well organized and well written. The topic is original and innovative.
Even if it is already clear, I suggest to describe “the aim” of the review at the end of the "introduction" paragraph.
Furthermore, to standardize the paragraphs, I suggest to remove the subtitle from paragraph 2.
Finally, I suggest defining the acronyms in the table legends.
Thanks for your suggestions. We added the aim of the review, removed the subtitle from paragraph 2 and defined the acronyms in the table legends, as you kindly suggested.
Reviewer 2 Report
This manuscript presents a comprehensive review of the role of post-genomic platforms, namely proteomics and metabolomics, in the identification of novel biomarkers in male hypogonadism. The authors have done a commendable job in discussing the application of these platforms in understanding the pathophysiological function of testosterone.
1. Originality and Significance: The manuscript provides a unique perspective on the use of proteomics and metabolomics in the study of male hypogonadism. It highlights the potential of these platforms in identifying novel, non-invasive markers of androgen activity and in shedding light on the different metabolic pathways associated with testosterone deficiency and replacement treatment.
2. Structure and Content: The manuscript is well-structured and the content is logically organized. The authors have provided a detailed background on the topic and have effectively discussed the application of proteomics and metabolomics in the study of male hypogonadism.
3. References: The manuscript includes a wide range of references, demonstrating a thorough review of the existing literature. However, it would be beneficial if the authors could include more recent studies to ensure the review is up-to-date.
4. Figures and Tables: The manuscript could benefit from the inclusion of more figures and tables to visually represent the key concepts and findings. This would enhance the readability of the manuscript and help readers better understand the content.
5. Language and Style: The manuscript is generally well-written. However, there are a few instances where the sentence structure could be improved for clarity. Additionally, the authors should ensure consistency in the use of terms throughout the manuscript.
6. Abstract and Introduction: The authors provide a comprehensive overview of the role of post-genomic platforms in identifying novel biomarkers in male hypogonadism. However, the introduction could benefit from a more detailed explanation of the significance of the study and its potential impact on the field.
7. Personalized -omic medicine in post-genomic era: The authors provide a detailed explanation of the evolution of precision medicine from genomics to proteomics and metabolomics. However, the section could benefit from a more explicit connection between these advancements and their application in the study of male hypogonadism.
8. Seminal proteomics in male hypogonadism: The authors present seminal proteomics as a promising tool for studying male hypogonadism. However, the discussion could benefit from a more detailed explanation of the specific proteins identified and their potential as biomarkers.
9. Sperm proteomics in hypogonadism: The authors discuss the role of testosterone in spermatogenesis and how sperm proteomics can reveal this. However, the section could be improved by providing more specific examples of the proteins identified and their roles in spermatogenesis.
10. Serum metabolomics in hypogonadism: The authors discuss the metabolic differences between functional and primary hypogonadism. However, the discussion could be improved by providing more specific examples of the metabolic pathways affected and how these differences could be used in the diagnosis and treatment of male hypogonadism.
11. Conclusions: The authors conclude by highlighting the potential of post-genomic platforms in the study of male hypogonadism. However, the conclusion could be strengthened by discussing the potential future directions of this research and its implications for the field.
Overall, the manuscript is well-written and presents a comprehensive review of the use of post-genomic platforms in the study of male hypogonadism. However, the authors could improve the manuscript by providing more specific examples and a more detailed discussion of the potential implications of their findings.
The manuscript is generally well-written and the English language used is clear and understandable. However, there are a few areas where improvements could be made:
1. Grammar and Sentence Structure: There are a few instances where the sentence structure could be improved for clarity. For example, in the sentence "The term ‘‘-omics’’ deals with the study of biological systems on a large scale," it would be clearer to say "The term ‘‘-omics’’ refers to the large-scale study of biological systems."
2. Use of Technical Terms: While the use of technical terms is appropriate for the subject matter, it would be helpful to define or explain some terms upon first use for the benefit of readers who may not be familiar with them.
3. Consistency: Ensure consistency in the use of terms throughout the manuscript. For example, if "male hypogonadism" is abbreviated as "MH" after first use, ensure this abbreviation is used consistently throughout the text.
4. Punctuation: There are a few instances where punctuation could be improved for clarity. For example, in the sentence "In this review the role of post-genomic platforms, namely proteomics and metabolomics, is analyzed with a specific insight on their application in the identification of novel biomarkers in MH and in the lighting on new knowledge into the mechanism of T action," a comma after "In this review" would improve readability.
5. Spelling: Ensure all words are spelled correctly. For example, "transaltional" should be "translational."
Overall, while the manuscript is generally well-written, these minor improvements could enhance its clarity and readability.
Author Response
We are grateful to the Reviewer for his/her observations.
This manuscript presents a comprehensive review of the role of post-genomic platforms, namely proteomics and metabolomics, in the identification of novel biomarkers in male hypogonadism. The authors have done a commendable job in discussing the application of these platforms in understanding the pathophysiological function of testosterone.
- Originality and Significance: The manuscript provides a unique perspective on the use of proteomics and metabolomics in the study of male hypogonadism. It highlights the potential of these platforms in identifying novel, non-invasive markers of androgen activity and in shedding light on the different metabolic pathways associated with testosterone deficiency and replacement treatment.
- Structure and Content: The manuscript is well-structured and the content is logically organized. The authors have provided a detailed background on the topic and have effectively discussed the application of proteomics and metabolomics in the study of male hypogonadism.
- References: The manuscript includes a wide range of references, demonstrating a thorough review of the existing literature. However, it would be beneficial if the authors could include more recent studies to ensure the review is up-to-date.
Thanks for your comments. We included a more recent study about the topic (connection between male hypogonadism and proteomics/metabolomics). Furthermore a recent new research has been done but we did not found any more recent study about this topic; however if you have in mind any specific study please give us the reference and we will include it in the review.
- Figures and Tables: The manuscript could benefit from the inclusion of more figures and tables to visually represent the key concepts and findings. This would enhance the readability of the manuscript and help readers better understand the content.
We added 2 figure reporting briefly the methodology used in the most modern proteomic studies of seminal plasma (Grande, G. 2019) and of sperm cells (Grande G, 2022).
- Language and Style: The manuscript is generally well-written. However, there are a few instances where the sentence structure could be improved for clarity. Additionally, the authors should ensure consistency in the use of terms throughout the manuscript.
A native English speaker revised the text for written English and approved the manuscript.
- Abstract and Introduction: The authors provide a comprehensive overview of the role of post-genomic platforms in identifying novel biomarkers in male hypogonadism. However, the introduction could benefit from a more detailed explanation of the significance of the study and its potential impact on the field.
We have added a sentence explaining the aim of the study and the potential impact on the field.
- Personalized -omic medicine in post-genomic era: The authors provide a detailed explanation of the evolution of precision medicine from genomics to proteomics and metabolomics. However, the section could benefit from a more explicit connection between these advancements and their application in the study of male hypogonadism.
We added some sentences explaining this connection.
- Seminal proteomics in male hypogonadism: The authors present seminal proteomics as a promising tool for studying male hypogonadism. However, the discussion could benefit from a more detailed explanation of the specific proteins identified and their potential as biomarkers.
We have discussed the specific proteins identified in seminal plasma as putative markers of male hypogonadism, as suggested.
- Sperm proteomics in hypogonadism: The authors discuss the role of testosterone in spermatogenesis and how sperm proteomics can reveal this. However, the section could be improved by providing more specific examples of the proteins identified and their roles in spermatogenesis.
We have discussed some proteins identified as differentially expressed in sperm cells reporting their roles in spermatogenesis.
- Serum metabolomics in hypogonadism: The authors discuss the metabolic differences between functional and primary hypogonadism. However, the discussion could be improved by providing more specific examples of the metabolic pathways affected and how these differences could be used in the diagnosis and treatment of male hypogonadism.
We added a sentence at the end of the paragraph underlining how these evidences may be used in the follow-up and treatment of patients with hypogonadism.
- Conclusions: The authors conclude by highlighting the potential of post-genomic platforms in the study of male hypogonadism. However, the conclusion could be strengthened by discussing the potential future directions of this research and its implications for the field.
We added a final sentence about the potential future directions of these post-genomic researches.
Overall, the manuscript is well-written and presents a comprehensive review of the use of post-genomic platforms in the study of male hypogonadism. However, the authors could improve the manuscript by providing more specific examples and a more detailed discussion of the potential implications of their findings.
We added some examples and discussed the potential implications
Comments on the Quality of English Language
The manuscript is generally well-written and the English language used is clear and understandable. However, there are a few areas where improvements could be made:
- Grammar and Sentence Structure: There are a few instances where the sentence structure could be improved for clarity. For example, in the sentence "The term ‘‘-omics’’ deals with the study of biological systems on a large scale," it would be clearer to say "The term ‘‘-omics’’ refers to the large-scale study of biological systems."
- Use of Technical Terms: While the use of technical terms is appropriate for the subject matter, it would be helpful to define or explain some terms upon first use for the benefit of readers who may not be familiar with them.
- Consistency: Ensure consistency in the use of terms throughout the manuscript. For example, if "male hypogonadism" is abbreviated as "MH" after first use, ensure this abbreviation is used consistently throughout the text.
- Punctuation: There are a few instances where punctuation could be improved for clarity. For example, in the sentence "In this review the role of post-genomic platforms, namely proteomics and metabolomics, is analyzed with a specific insight on their application in the identification of novel biomarkers in MH and in the lighting on new knowledge into the mechanism of T action," a comma after "In this review" would improve readability.
- Spelling: Ensure all words are spelled correctly. For example, "transaltional" should be "translational."
Overall, while the manuscript is generally well-written, these minor improvements could enhance its clarity and readability.
The text has been amended to take in along with your valuable suggestions. At the end of the revision, a native English speaker revised the text for written English and approved the manuscript.
Round 2
Reviewer 2 Report
Accept in present form